The effect of age and sexual maturation on thermal preferences of honey bee drones

Czekońska Krystyna
Łopuch Sylwia Sylwia.Lopuch@urk.edu.pl
Department of Zoology and Animal Welfare, University of Agriculture in Krakow , Krakow , Poland
Negri Ilaria
Electronic publication date: 2022 Jun 29
Publication date: 2022
Volume: 10
Electronic Location ID: e13494
Received 2021 Dec 14; Accepted 2022 May 4
Copyright: ©2022 Czekońska and Łopuch
Copyright year: 2022
Copyright holder: Czekońska and Łopuch
License: This is an open access article distributed under the terms of the Creative Commons Attribution License, which permits unrestricted use, distribution, reproduction and adaptation in any medium and for any purpose provided that it is properly attributed. For attribution, the original author(s), title, publication source (PeerJ) and either DOI or URL of the article must be cited.
License URL: https://creativecommons.org/licenses/by/4.0/

Keywords: Apis mellifera carnica, Drone age, Temperature, Mucous gland

Funding: The authors received no specific funding for this work.

==============================
The thermal preferences of Apis mellifera carnica drones (male individuals) are poorly understood, though their reproductive quality affects the quality of the inseminated queen and the whole honey bee colony. Therefore, the aim of this study was to determine the thermal preferences of individual drones according to their age and sexual maturity. Drones at the ages of 1, 5, 10, 15, 20 and 25 days were tested. The drones were placed on a platform in a temperature gradient in the range 20 °C and 46 °C. The thermal preferences of the drones were measured with the use of a thermal-imaging camera. Drones significantly differed with their choice of a preferred temperature. The one-day-old and the 25-day-old drones preferred the lowest temperatures. A slightly higher temperature was preferred by the 5-day-old drones, and the highest temperature was chosen by the drones at the ages of 10, 15, and 20 days. The changes in the thermal preferences of drones correspond to physiological changes occurring with age and connected with the rate of sexual maturation.

Introduction

The development and functioning of a honey bee colony strongly depends on the ambient temperature that determines the activities of a colony and each individual bee, and also determines the course of metabolic processes (Abou-Shaara et al., 2017; Cook et al., 2021). Inside the nest, the temperature is actively regulated by the workers and adapted to the colony’s needs (Stabentheiner, Kovac & Brodschneider, 2010; Stabentheiner et al., 2021). In the centre of the nest, in the place of the brood rearing, the temperature is maintained at a level of 33–36 °C (Kleinhenz et al., 2003; Abou-Shaara et al., 2017). As the distance from the nest centre increases, the temperature decreases. On the edges of a comb, in the area of drone brood rearing, the temperature falls by about 1.5 °C, whereas in other places in the comb, where honey is collected, the temperature can be maintained at below 32 °C (Kronenberg & Heller, 1982; Stabentheiner, Kovac & Brodschneider, 2010; Scheiner et al., 2013).

The temperature of a honey bee nest meets the thermal requirements of all the bees, but this does not mean that particular individuals always stay at their preferred temperature. The research has so far focused mainly on the range of the thermal functioning of individual bees, and their behaviour in different temperatures (Free & Spencer-Booth, 1960; Levin & Collison, 1990; Hrassnigg & Crailsheim, 2005; Kovac et al., 2014). Additionally, the research has mostly concentrated on workers (Grodzicki & Caputa, 2005; Grodzicki & Caputa, 2014; Grodzicki, Piechowicz & Caputa, 2020), and only sporadically on male individuals called drones (Ohtani & Fukuda, 1977; Crailsheim et al., 1999; Kovac, Stabentheiner & Brodschneider, 2009), which considerably differ from workers anatomically, morphologically, physiologically and behaviourally (Hrassnigg & Crailsheim, 2005). Workers develop from fertilised eggs and their development takes 21 days, whereas drones develop from unfertilised eggs and their development takes 24 days (Hrassnigg & Crailsheim, 2005). Drones are more sensitive to temperatures and stay active in a narrower temperature range of 14–40 °C, whereas workers are active in a temperature range of 10–46 °C (Free & Spencer-Booth, 1960; Cahill & Lustick, 1976; Heinrich, 1979; Stürup et al., 2013). In a honey bee colony, drones are responsible for producing and transmitting semen to the queen during a mating flight. The reproductive quality of drones influences the quality of the inseminated queens, which, in turn, directly affects the quality of the whole colony (Boes, 2010; Brutscher, Baer & Niño, 2019; Rangel & Fisher, 2019). However, there is very little data on the thermal preferences of drones depending on their age (Mindt, 1962; Crailsheim et al., 1999).

In a honey bee colony, drones occur seasonally during the reproductive period (i.e., swarming) and they usually constitute from 5% to 10% of the population of adult bees (Boes, 2010). In the nest, drones stay in different temperature zones according to their age (Örösi Pál, 1959; Free, 1967; Ohtani & Fukuda, 1977; Crailsheim et al., 1999; Kovac, Stabentheiner & Brodschneider, 2009). It is suggested that young drones stay more frequently in the brood rearing area, where the temperature is higher, whereas older drones leave this zone (Free, 1957; Ohtani & Fukuda, 1977; Harrison, 1987; Abd Al-Fattah, El-Shemy & El-Masarawy, 2016). They leave the nest only when the ambient temperature is above 20 °C (Drescher, 1969; Cahill & Lustick, 1976).

In honey bee drones, the spermatogenesis process starts at the larva stage and is completed at the end of the pupa stage (Bishop, 1920; Lago et al., 2020). After eclose, drones need some time to mature for copulation. They reach sexual maturity (i.e., capability to mate with the queen) at the age of 10–16 days (Woyke & Ruttner, 1958; Rhodes, 2002). During the first days of life of a drone as an imago, spermatozoa pass from testicles to seminal vesicles where they are stored until copulation (Bishop, 1920; Hayashi & Satoh, 2019). Directly after the eclose of a drone, the process of the production of mucus and the maturity of mucous glands also begins (Moors et al., 2005), at the end of which drones are ready to copulate with a queen (Cruz-Landim & Dallacqua, 2005). Drones copulate only once, most frequently at the age of 15–23 days, with the average at 21 days (Couvillon et al., 2010).

Reproductive quality and the rate of maturation to copulation depend, to a large extent, on the nutritional status and temperature at which the drones settle during their postembryonic development (Hrassnigg & Crailsheim, 2005; Czekońska, Chuda-Mickiewicz & Chorbiński, 2013; Czekońska, Chuda-Mickiewicz & Samborski, 2015; Szentgyörgyi, Czekońska & Tofilski, 2017; Szentgyörgyi, Czekońska & Tofilski, 2018), and the temperature in which drones stay until copulation (Stürup et al., 2013; Rangel & Fisher, 2019). In the nest, some drones can stay at the range of their preferred temperatures, whereas others will only stay within the range of tolerated temperatures due to temperatures fluctuations (Abou-Shaara et al., 2017; Cook et al., 2021) or the presence of parasites (Duay, De Jong & Engels, 2002).

Gathering of young drones in the warmest part of the nest is explained by the higher temperature (Free, 1967; Örösi Pál, 1959; Ohtani & Fukuda, 1977) or the presence of large number of nurse bees, which can feed or groom the young drones (Crailsheim et al., 1999; Goins & Schneider, 2013). Kovac, Stabentheiner & Brodschneider (2009) reported that the distribution of young drones in the nest is more differentiated when the number of drones at the age of three to seven days old does not exceed 36% in warmer brood area. It can be, therefore, expected that the drone’s distribution may depend on their individual thermal preferences. We suggest that the thermal preferences of the drones can depend on the degree of their maturity to copulate with a queen. Our hypothesis might explain the drones’ reproductive sensitivity to temperature depending on age (Czekońska, Chuda-Mickiewicz & Chorbiński, 2013; Stürup et al., 2013; Rangel & Fisher, 2019). We suggest that sexually immature young drones, whose spermatozoa have passed from their testicles to their seminal vesicles, may differ in their thermal preferences from sexually mature older drones preparing for their mating flight and transferring semen to the queen. In order to explain this, we performed this study that aimed to know the thermal preferences of drones according to their age and the degree of their maturity to mating.

Material and Methods

Drones

The study was conducted on Apis mellifera carnica drones at known ages from three colonies in the experimental apiary belonging to the University of Agriculture in Krakow located at Garlica Murowana. Drones were reared from May to June 2020. Drones at the same age came from combs with drone brood that were placed in an incubator 24 h before the expected eclose, at a temperature of 34 °C and a relative humidity of 50–60%. Directly after eclose, the drones were collected from each comb (n = 3) and each colony (n = 3), and placed in separate wooden cages (8 × 10 × 5 cm). The cages with drones were placed in three strong unrelated colonies located next to the laboratory of the University of Agriculture in Krakow, and three strong colonies in the experimental apiary as reserves. Approximately 1,800 drones were collected. Previous research indicated that the nest environment has little effect on adult drone physiology and had no effect on the life span and survival of the adult drones (Stürup et al., 2013; Czekońska, Szentgyörgyi & Tofilski, 2019). In the hives, cages with drones were placed between two peripheral combs. In each cage, one wall was made of an excluder to ensure the constant and free access of workers to drones. The location of the cages within the hives allowed easy access to them without disturbing the colony’s work. In each colony, there were a maximum of six cages, with 50 drones in each one.

The temperature platform

The base of the temperature platform was an aluminium sheet approximately 1 m ×1 m (Fig. 1). The sheet was insulated by four mm polyethylene foam on the underside. On one edge of the sheet, a heating strip with a temperature regulator ESCO ES10 was mounted. On the opposite edge, a cooling strip made of fans connected with a modular power supply LED PB021505 (300W, 12V, 25A) was attached. Such a construction facilitated the maintaining of the platform at the required temperature gradient within the range 20 °C to 46 °C from a cooling edge to a heating edge. On the top side, the sheet was divided by aluminium T-profiles (about one mm width, 20 mm height and 1,000 mm length) into 14 sectors. Each sector was divided by aluminium U-profiles (about 10 mm width, 0.8 mm height and 1,000 mm length) on two measurement paths in which drones were placed individually. Along each measurement path, the surface of the aluminium sheet was covered by black insulating tape, preventing light reflection. The sides of each sector were closed off by carton dividers. The temperature platform was covered on the top by a cellophane sheet to prevent drones from moving out of their sectors and to help maintain the stability of the thermal gradient.

The experiment

The experiment was conducted in the laboratory between 12:00 and 15:00 keeping the order of repetitions performed. Initially, measurements were taken in a darkened room where the windows were covered by blinds. However, because diffused daylight was causing increased drone activity, the measurements were taken in darkness in a tent made of black foil, in conditions similar to those prevailing in the nest. Inside the tent, all work was performed in the red light of an LED lamp. The experiment was conducted in a room temperature of 24 °C. Drones were placed in the measurement paths from the edge of the platform where the temperature was lower.

Figure 1 The temperature platform: (A) front view, (B) top view.

Before each test of thermal preferences, cages with drones at a particular age were placed in the incubator at a temperature of 30 °C for 15 min to standardise experimental conditions. One cage in three containing drones from a given comb was used in the testing. In the first place, drones kept in three colonies located next to the laboratory were used for testing. Drones from the other cages were reserves, and they were used for testing as some drones from the first cage were lost. During testing, the drones were not fed.

Drones in the first day of life were tested in diffuse daylight in a darkened room and in darkness in a tent, whereas older drones were tested only in darkness. After placing the drones in measurement paths on the temperature platform, they were undisturbed for 10 min. The preliminary data showed that during this time, drones calmed dawn and stayed in the chosen place (Fig. 2). Then, an image of the place on the temperature platform, where each of the tested drones was staying, was recorded with the use of a FLIR E8 thermal-imaging camera in order to determine the temperature chosen by each drone. The temperature in which a drone stayed was read from the surface of the platform covered by black insulating tape with an accuracy of up to 0.1 °C using FLIR Tools software, version 5.13 (Fig. 2). The emissivity value was settled at 0.95 according to the instruction of the camera in order to correct for reflection of ambient radiation. The images were taken at the same distance from the platform. The attenuation of infrared radiation by the cellophane sheet covering the platform was also corrected during the evaluation setting in the software to a value allowing to compensate that change, which was 0.1 °C. After the test was completed, the drones were returned to their cages and to the foster colonies where they were kept until the next measurements. At any one time, up to 28 drones were being tested, one drone in each measurement path. A total of 1,154 images were taken.

Figure 2 The image of the place on the temperature platform preferred by a drone taken with a thermal-imaging camera: (A) an infrared photograph (B) a standard photograph.

The thermal preferences were assessed in drones at the ages of 1, 5, 10, 15, 20 and 25 days referred to later in this paper as D1, D5, D10, D15, D20, and D25. Individual drones at the ages of 10, 15, and 20 days, which stayed at the extreme ends of the gradient temperature (20 °C and >44 °C), were collected to analyse the degree of their sexual maturity. For this purpose, 10 drones were selected from each age group and each end of the gradient temperature. The sexual maturity of the drones was assessed based on the degree of their mucous-gland development (Moors et al., 2005). After preparing, the reproductive organs were placed on a microscope slide and the degree of the filling of their mucous glands with secretion was evaluated, using a stereoscopic microscope Delta Optical SZ-453T. The categories of mucous-gland development were established based on the degree of their filling with secretion, as well as the colour and thickness of the secretion.

Statistical analysis

Parametric one-way ANOVA was used to evaluate the influence of light on the thermal preferences of the drones tested in diffused daylight or in darkness. A comparison of the thermal preferences of one-day-old drones was performed based on 191 images, 88 being made in diffused daylight and 103 in darkness.

During the analysis of the collected images, 38 were rejected due to the behaviour of the drones in a temperature gradient above 44 °C (i.e., the drones were settling on carton dividers that closed the end of each sector). Finally, 1,116 images were used in the analysis. Due to the absence of a normal distribution examined with the use of the Kolmogorov–Smirnov test, the thermal preferences of the drones according to their age were assessed using a non-parametric Kruskal-Wallis (KW) test. Multiple (post-hoc) comparisons among age groups of the drones were performed using Dunn’s test. Fisher’s exact test was used to compare frequencies of the drones differing with their mucous-gland development due to small number of groups. All statistical analyses were performed using Statistica software, version 13.

Results

The effect of light on the thermal preferences of the drones

The light had no significant influence on the thermal preferences of the drones (ANOVA: F(1, 189) = 1.751, p = 0.187).

The effect of age on the thermal preferences of the drones

Drones significantly varied in their thermal preferences depending to their age (KW test: H = 102.23, df = 5, n = 1116, p < 0.001; Fig. 3; Table 1). Drones from groups D1 and D25 preferred the lowest temperatures (mean ± SE: 30.9 ± 0.29 °C and 32.7 ± 0.52 °C, respectively). A slightly higher temperature (mean ± SE: 33.1 ± 0.46 °C) was preferred by D5 drones, and the highest temperature (mean ± SE: 35.4 ± 0.36 °C, 35.0 ± 0.57 °C and 35.1 ± 0.44 °C) was chosen by the D10, D15 and D20 drones. Drones also differed with the range of preferred temperatures. Young drones (D1 and D5) stayed in the temperature gradient between 20 °C and 40 °C, whereas older ones (D10, D15, D20 and D25) preferred a higher range of temperatures—between 24 °C and 44 °C.

Figure 3 Median ± IQR (interquartile range)—the preferred temperature of drones at different ages (the same letters indicate the lack of significant differences at p > 0.05).

The effect of the development of the mucous glands on the thermal preferences of the drones

The evaluation of the colour and degree of development of the mucous glands of drones at the same age, which stayed in an extreme gradient of temperatures, showed anatomical and physiological differences in their rate of maturation (Table 2, Figs. 4A–4D). Drones that preferred a lower temperature (24–26 °C) showed more advanced development of the mucous glands compared with drones at the same age choosing a higher temperature (42–44 °C; Table 2). Differences in the degree of filling of the glands with the secretion of D10, D15 and D20 drones, staying in an extreme gradient of temperatures, were significant (Fisher’s Exact test: p = 0.008, n = 10) in each age group (Table 2).

Table 1 The significance (p values, post-hoc Dunn’s test) of the differences between the preferred temperature of drones at different ages.

Age (days)	1	5	10	15	20	25	
1		0.013	<0.001	<0.001	<0.001	0.351	
5	0.013		0.017	0.510	0.235	1.000	
10	<0.001	0.017		1.000	1.000	<0.001	
15	<0.001	0.510	1.000		1.000	0.037	
20	<0.001	0.235	1.000	1.000		0.009	
25	0.351	1.000	<0.001	0.037	0.009		

Table 2 Differences in development of the mucous glands of drones preferring the extreme ends of the gradient temperature.

Degree of development
of the mucous glands of drones	Temperature gradient	Fisher’s exact test	
	Low
(24–26 °C)	High
(42–44 °C)		
10-days-old	Completely filled with pearl-white, sparse secretion (n = 5)	Partly filled with white secretion (n = 5)	p = 0.008	
15-days-old	Completely filled with white and thick secretion (n = 5)	Completely filled with pearl-white secretion of different thickness (n = 5)	p = 0.008	
20-days-old	Completely filled with white and thick secretion, which is also in the distal parts of the reproductive tract (n = 5)	Completely filled with white and thick secretion (n = 5)	p = 0.008	

Figure 4 Stages of development of mucous glands of drones: (A) partly filled with secretion, (B) completely filled with sparse secretion, (C) completely filled with thick secretion, (D) secretion of the mucous glands in the distal parts of the reproductive tract.

Discussion

The results of the present study show that the thermal preferences of drones change with their age. Younger drones at the age of one, five and the oldest ones at the age 25 days prefer lower temperatures (median: 30.6–34.3 °C), whereas older drones at the age of 10-20 days prefer higher temperatures (median: 33.8–35.6 °C). The range of preferred temperatures also changes with drone age, as younger drones (D1 and D5) preferred a lower range of temperatures (20–40 °C), whereas older ones (D10, D15, D20 and D25) preferred a higher range of temperatures (24–44 °C). The changes in the thermal preferences correspond to the subsequent development stages of the drones. The type of light does not affect the thermal preferences of the drones, but diffuse daylight stimulates their motor activity.

The previous studies showed that younger drones were mostly concentrated in the warmer brood area, whereas older drones were on the colder peripheral non-brood areas (Free, 1957; Örösi Pál, 1959; Ohtani & Fukuda, 1977; Harrison, 1987; Crailsheim et al., 1999; Goins & Schneider, 2013; Abd Al-Fattah, El-Shemy & El-Masarawy, 2016). However, distribution of drones depending on their age is not homogenous (Kovac, Stabentheiner & Brodschneider, 2009), and may depend on other factors, including the degree of nutrition or nurse bees care (Mindt, 1962; Crailsheim et al., 1999; Goins & Schneider, 2013). Our results indicate that distribution of drones in the nest may also depend on their thermal preferences. The one-day-old, five-day-old and the 25-day-old drones preferred lower temperatures in the range of optimal temperatures, whereas 10-, 15-, and 20-day-old drones preparing to mate preferred higher temperatures. The one-day-old drones may choose lower temperatures due to a lack of completely developed endothermy (Kovac, Stabentheiner & Brodschneider, 2009), whereas the oldest drones (>25 days) may move to the peripheral storage area where they feed themselves from open cells of honey (Jaycox, 1961). The methodical differences can be responsible for the observed discrepancies between our and previous results in relation to older drones preferring higher temperatures. In our study, the drones were tested individually on the temperature platform, whereas in other studies drones were investigated mostly on the combs in hives (ordinary or observation) in the presence of other bees. Our study for the first time examined individual thermal preferences of drones separated from the effects of nestmates, mostly nurse bees, and local temperatures prevailing on different parts of the combs.

Our results indicate that changes in the thermal preferences of drones occurring with age correspond to anatomical and physiological changes reported in other studies connected with their maturation (Bishop, 1920; Moors et al., 2005; Hayashi & Satoh, 2019; Lago et al., 2020). The dimensions of the reproductive organs change along with the age of the maturing drones (Czekońska, Chuda-Mickiewicz & Chorbiński, 2013; Metz & Tarpy, 2019). We suggest that young drones in the age groups of D1 and D5 prefer lower temperatures until the spermatozoa are in the testicles. After passing sperms from the testicles to the seminal vesicles, which takes place between the third and the eighth day of the drone’s life (Bishop, 1920; Hayashi & Satoh, 2019; Metz & Tarpy, 2019), they prefer higher temperatures. Our results indicate that preferences of 10-day-old drones, where the spermatozoa are in the seminal vesicles, for a higher temperature may be linked with their physiological and physical preparation for the next stage of sexual maturation for the mating flight and copulation with a queen. The changes in the thermal preferences of ready to mate drones can result not only from the need to eat honey, but also from physiological causes as evidenced by the lower temperature chosen by 25-day-old drones.

During drone maturation, the mucous glands develop, filling with secretion, which gradually changes colour from transparent to milky-white (Moors et al., 2005; Czekońska, Chuda-Mickiewicz & Chorbiński, 2013; Metz & Tarpy, 2019). The production of this secretion, which plays an important role during the transfer of semen to the reproductive tract of the queen (Bishop, 1920; Colonello & Hartfelder, 2003; Colonello-Frattini & Hartfelder, 2009), is completed before drones reach sexual maturity (Cruz-Landim & Dallacqua, 2005). At this time, drones begin to show a greater tolerance to higher temperatures (Stürup et al., 2013). This indicates that drones may need higher temperatures in their final stage of maturation, and therefore, they prefer them. The results of the present study clearly indicate that the drones—at the ages of 10, 15, and 20 days—chose the highest temperatures.

In our opinion, the variation in the thermal preferences of drones at the same age mainly result from different rates of maturity, as differences in the degree of mucous-gland development of drones staying at extreme temperatures indicate, and were observed mainly in the age groups D10 and D15. The evaluation of the mucous glands appears to indicate that slower developing drones at the ages of 10 and 15 days preferred higher temperatures than the same-aged drones, which chose lower temperatures. It is likely that the thermal preferences of drones at the age from 10 to 20 days can be associated with the biochemical changes occurring in the composition of the mucous gland secretion (Colonello & Hartfelder, 2003; Cruz-Landim & Dallacqua, 2005). Morphological and histological changes of the mucous glands are observed until the ninth day of life (Bishop, 1920; Colonello & Hartfelder, 2003). The research indicates that lower temperatures can retard drone maturity (Jaycox, 1961) and influence the slowing down of testes atrophy (Stoian, Papuc & Petrescu-Mag, 2020). Drones at different stages of development are exposed to fluctuating temperatures in the nest, to a greater or lesser extent, which may retard or accelerate their sexual maturation. The link between the thermal preferences of drones and the degree of their sexual development needs further research, which makes it difficult, because of the lack of the possibility to the maintain reproductive organs of drones without them being damaged or without changes in filling the mucous glands with secretion as a result of freezing (Carreck et al., 2013).

Drones at the same age can differ in their rate of maturation due to various causes, not only fluctuating temperatures in the nest, but also the maternity colony, nutritional status, or body mass (Jaycox, 1961; Ohtani & Fukuda, 1977; Crailsheim et al., 1999; Boes, 2010; Czekońska, Chuda-Mickiewicz & Samborski, 2015; Czekońska, Szentgyörgyi & Tofilski, 2019; Szentgyörgyi, Czekońska & Tofilski, 2016; Szentgyörgyi, Czekońska & Tofilski, 2017). The indicated differences can influence on the distribution of drones in the nest (Ohtani & Fukuda, 1977; Kovac, Stabentheiner & Brodschneider, 2009). The drones in our study came from three different colonies that could also impact on their rate of maturation. It is possible that drones at the same age that mature faster move to a zone with a higher temperature earlier (Stürup et al., 2013).

In conclusion, our results indicate that the thermal preferences of drones correspond to physiological changes occurring with their age, and are consistent with the degree of their maturity to copulation. Further research is needed to better understand the effects of thermal conditions on drone reproductive quality.

Supplemental Information

Supplemental Information 1 Raw data: Temperatures preferred by drones of different age

Click here for additional data file.

We would like to thank Prof. Adam Tofilski for constructing the temperature platform. We also thank the Reviewers for their suggestions and comments.

Additional Information and Declarations

Competing Interests

Author Contributions

Data Availability

The authors declare there are no competing interests.

Krystyna Czekońska conceived and designed the experiments, performed the experiments, analyzed the data, prepared figures and/or tables, authored and reviewed drafts of the article, and approved the final draft.

Sylwia Łopuch performed the experiments, authored and reviewed drafts of the article, and approved the final draft.

The following information was supplied regarding data availability:

The raw measurements are available in the Supplementary File.

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
