# Peer review of "The effect of age and sexual maturation on thermal preferences of honey bee drones"

_PeerJ, doi:10.7717/peerj.13494_

## Round 0.1 · original submission · Major Revisions

In my opinion, this article needs substantial improvements prior to publication.

The major concern arises from conflicting results presented in Table 1 and in the boxplot of Figure 3, as in Fig.3 differences between the preferred temperature of drones of group D5 and group D25 are significant (p < 0.05), while in Table 1 p = 1. This of course affects the discussion.

Also, both reviewers expressed concern about the clarity of materials and methods that may affect reproducibility.

The whole text must be carefully revised also because some sentences appear confusing (e.g., the discrepancies in the results declared by the authors) or redundant. Some references to figures/tables in the text are wrong.

More emphasis to the part concerning the evaluation of the colour and degree of development of the mucous glands should be given. Table 2 is not very informative as it is and I suggest adding chi-square results and values of significance.

·

Basic reporting

I have line notes and a section-by-section commentary that more clearly delineates my opinions here, but broadly speaking, the authors clearly present the background literature, methods, and data using professional English, although some word use and grammatical suggestions are made. The literature review is thorough with context provided, although I have some recommendations about streamlining and improving the logic of the hypothesis. The paper is self-contained and represents a complete experimental unit with a purpose and impact. The figures and tables are basic, but perfectly adequate for the experiment. There appears to be some data missing pertaining to mucous gland development that I would like to see.

Experimental design

This paper meets the aims and scope of the journal. The research question is well-defined (with the above caveats) and fills a knowledge gap. The design of the experiment is adequate to the question, although arguments about statistical power can be made. I don't strictly think this is a barrier to publication, but is something of which to be aware. Methods are broadly replicable, although I make some recommendations about how to improve this, given that a bespoke apparatus was constructed to perform the described tests.

Validity of the findings

The conclusions are well-stated and largely clear. As mentioned above, some of the data seems to be missing or I'm misunderstanding. Where the experiment does have noticeable pitfalls, these are appropriately called out and analyzed.

Additional comments

This is a very interesting paper, presented in a straightforward and thoughtful manner. I have some items for improvement, but overall I would recommend that this paper be considered for publication after some revisions are performed.

Introduction:
The authors present a fairly detailed literature review that decently summarizes the state of research with regard to drone temperature sensitivity and nest homeostasis more generally. The case for understanding an individual-level variation in age-based preference for temperature is intriguing and has interesting extensions for understanding organismal impacts on sexual development, namely spermiogenesis and spermatozoa transfer to the seminal vesicles. The logic presented in the introduction, however, needs some work to be convincing. There are additionally some elements that appear repetitive or redundant (or I am not understanding the distinctions made). It is unfortunately unclear from the narrative what the exact niche the paper is trying to fill. I therefore recommend some refocusing of the introduction with specific line items noted below.

45: “While drones considerably differ…” This appears to be an incomplete sentence.

63: “A drone is an exceptional example…” I’m not clear on what this sentence is trying to say. It seems redundant with the subsequent review.

64: The “spermatogenesis process…”

66: “Process of gonadal…” It might be wise to define sexual maturity here. If you mean capability to ejaculate, that is a different time point than spermatozoa migration to the seminal vesicles.

78: “In the nest, due to spatial…” I’m not clear on the intended meaning of this sentence. Do you mean that some drones are unable to find their “preferred” temperature? If so, how is this established?

81: “In line with previous…” It would be better to just tell us why nurse bees and young drones should be associated. I assume feeding, but

84: “Previous research…” It seems that this review is somewhat redundant with 58-62 and contradictory with the statement in 53-54. I get that the common scientific trope is, “X is understudied,” but try not to prove yourself wrong in the literature review. I think that you mean that individual variance in age-based temperature variance is an unknown, but you need to emphasize this more.

86: “However, it has been recently reported…” Where please? Also, the three sentences here to line 90 present a logic chain that I’m sympathetic to, but not following. Here’s my interpretation: 1) Age distribution of drones over brood and non-brood areas is different. 2) About 36% of young drones hang out over brood. 3) Therefore, there is individual preference? I don’t think that 3 arises from 1 and 2. Given that this is the rationale of the paper, I need a more convincing logical thread here.

95: “…whose sperms…” I would use spermatozoa instead.

99: “…according to their age and degree of their maturity to mating.” Here it sounds like you’re suggesting a decoupling of age and sexual maturity. This is very interesting and something that might be worth further exploration, especially if the introduction is refocused as suggested above.

Materials & Methods:
The materials and methods appear adequate for the experimental design, the statistics are reasonable, and any potential shortcomings that I can see are already noted in the text for caveats. The only thing I might wish for is a photo of the testing apparatus, which, while well-described, remains somewhat difficult to visualize. I’m also not capable of evaluating the measurement tolerance (precision particularly) of the camera, so we must trust the authors that the technology is “proven” as it were. I’m not sure there’s a corrective action necessary, but it’s a thought. I have a few notes below.

155: “The emissivity value was” settled…

158: “…was also corrected for during evaluation setting…” This sentence could use some revision for English language clarity.

158-161: The sentence about transferring files to a computer is probably not necessary.

161-162: This sentence implies that the statistics were performed manually in Excel. As this is unlikely to be the case, I assume this statement is redundant with the subsequent statistical methods provided.

162: “…drones went back…” presumably the drones were returned to their cages and foster colonies and did not go there of their own accord.


Results:
Broadly, the results are clearly presented and easy to interpret. I have some concerns about the final, Chi2 square analysis in that it seems to lack the clarity of presentation necessary to properly interpret. This particular analysis further is not represented in the provided data and appears to be incomplete; the reporting seems to only encompass one of the three tested groups. Finally, an n of 5 is likely far too low for a chi square test and an alternative should likely be performed (e.g. Fisher’s exact test). I would therefore recommend some increased detail and alternative analyses in these areas.

182: I would call the procedure, “Dunn’s test.”

196-200: I rebel in this in my own writing, but it would be helpful, if somewhat redundant, for summary statistics to be provided in the descriptors of the drone group temperature preferences. The sample number is fine, but I would rather have means+standard errors or medians+ IQRs. The sample numbers would be better placed in the statistical table, or as annotations to the accompanying figure.

201-203: since statistically this is a further description of the analyses above, this statement should be within the prior paragraph in my view.

204-209: this appears to be the most novel part of the manuscript, and it unfortunately gets the least page time. I have some concerns that the replication is low (5 per group out of ~1100 drones?). Also, there’s no data table associated with this, or I’m failing to understand what Table 2 is trying to say. My thinking is that the result is suggesting that within age groups, those drones that are at the warmer or colder end show differential qualitative maturity scores based on mucous glands. If so, there should be a score associated with each drone, which unfortunately is not presented in the provided data. I’m not certain where the Chi2 test is pulling from here. My recommendation is to spend more time on this interesting result and flesh out the analysis and presentation further.

Discussion:
The discussion clearly presents the results and contextualizes the results with the introduction. I have a few notes on focusing, but any other notes are largely matters of taste and style.

213: “Young drones of…” The data show a non-linear trend, but this opening statement suggests a linear trend, consequently misidentifying the oldest age group as being, “younger.” This should be revised.

229: What discrepancies in results? I’m not clear on the issue.

238: what is “inter alia?”

240: I still prefer “spermatozoa” over sperms. And the spermatozoa are present in the testes at the age groups mentioned, and will migrate to the seminal vesicles, as the authors note in the subsequent sentence.

243-246: This is the most interesting finding of the paper. Highlight this and expand if plausible. Support this statement with improved statistical analyses and reporting.

270-272: I’m not sure what this sentence is trying to say.


References:
The titling of the references need to be made consistent. Some are capitalized (e.g.: 314-316) and some are not (e.g.: 317-319).

Table 1: Please include the test performed in the caption. Also include the test statistic and degrees of freedom in the table for full clarity. I know these tables are somewhat procedural, but it does help in reader interpretation. You might also include significance groups directly in the table.
Table 2: This table should have detailed statistics added in lieu of a Chi2 figure.

Figure 3: Correct “Mediana” to “Median.”

Reviewer 2 ·

Basic reporting

I found the manuscript interesting but occasionally confusing and needs some revisions, plus a thorough review of the English language and form.
Title: The title is representative of only part of the work. The manuscript studies the preference in temperature of drones but also the grade of maturation of the sexual glands at different temperatures. I suggest improving the title to cover both aspects.
Abstract: please provide the name of the honey bee sub-species and make the aim of this manuscript more explicit.
Keywords: add Apis mellifera carnica
Introduction: Add the context and a brief description of Apis mellifera carnica to better understand the work to non-specialists.
line 30: add other references if available (e.g., https://doi.org/10.1093/jee/toab023)
lines 45-46: “While …behaviourally” something is clearly missing here.
line 46: explain the drone life cycle (the development of drones from the larvae stage is longer than queen and workers).
line 55: add a comma after i.e.
line 63: specifies that drones are haploid.
lines 65 and 69: please substitute the “emergence” with eclose, here and elsewhere.
lines 90-99: Are these the aims of work? If yes, improve the text to make it more explicit and clearer.

Experimental design

Materials and methods: Materials and methods look quite confusing to me, possibly hampering the reproducibility of the experiment. I would suggest including a drawing of the experimental setup and possibly of the experimental design.
Drones:
line104: where and when did you collect the colonies?
line105: how many broods did you collect for the colony?
Line 107: how many drones did you collect?
Line 108: is there one single cage per colony?
Lines108-111: Why did you put the drones in a different colony?
The temperature platform
Please improve the paragraph with more details and add the photo / drawing of the temperature platform.
Line 119: Which material was used to insulate?
Statistical analysis
This paragraph can be improved with more details (for example, the information at lines 187-188 should be included here).

Validity of the findings

Results
This paragraph should only include the result of the experiment without hastening the discussion.
Lines 187-188: please include this part in M&M (statistical analysis).
Lines 190-192: This belongs to ‘discussion’.
Lines 193-195: This belongs to M&M (statistical analysis).
Lines196-200: This section puzzles me. It looks like the results at Table 1 and Fig 3 (plot with median and CI) refer to the same results, but the information provided by the numbers would suggest otherwise (in Table 1 D5 vs D25 is not significant, whereas D5 vs 25 in Fig 3 is significantly different). Please explain or modify if necessary.
Discussion
I can not comment the discussion because the results are not adequately clear.

Please check all the references of the figures in the main text.
Figures:
• Figures have been uploaded as single figures, but I believe they were meant to be arranged as combined figures.
• Photos need a size reference.
• You could also include some arrows pointing at significant details (the distal part of the reproductive tract).

---

## Round 0.2 · Minor Revisions

The manuscript has been substantially improved and only minor changes are requested prior to publication.

I suggest the authors carefully follow the reviewer's comments.

Reviewer 2 ·

Basic reporting

I suggest providing the actual temperature values and range when you mention low and high temperatures in the results and discussion sections, and table 2.

Figures:
• Figure 1 You could also include some arrows pointing at the main parts of the temperature platform and, if possible, add a new photo with a view from the top.
• Figure 4 It still needs the bar scale reference.

Experimental design

No further comments

Validity of the findings

No further comments

Additional comments

No further comments

---

## Round 0.3 · accepted · Accept

The revisions are effective in addressing the remaining concerns. In my opinion, the article is now ready to be published.